

**Technical note: A low-cost albedometer for snow and ice measurements –**
**Theoretical results and application on a tropical mountain in Bolivia**
Thomas Condom[1*], Marie Dumont[2], Lise Mourre[1], Jean Emmanuel Sicart[1], Antoine
Rabatel[1], Alessandra Viani[1], Alvaro Soruco[3]
[1] Université de Grenoble Alpes, IRD, CNRS, Grenoble-INP, IGE (UMR5001), F-
38000 Grenoble, France
[2] Météo-France, CNRS, CNRM-GAME/CEN (UMR3589), Grenoble, France
[3] UMSA, Instituto de Geológicas y del Medio Ambiente, La Paz, Bolivia
*Corresponding author: thomas.condom@ird.fr



## Abstract

This study presents a new instrument called a low-cost albedometer (LCA) composed
of two illuminance sensors that are used to measure *in-situ* incident and reflected
illuminance values on a daily timescale. The ratio between reflected *vs.* incident
illuminances is called the *albedo index* and can be compared with actual albedo values.
Due to the shape of the sensor, the direct radiation for zenith angles ranging from 55°
to 90° is not measured. The spectral response of the LCA varies with the solar
irradiance wavelengths within the range 0.26 to 1.195 µm, and the LCA detects 85%
of the total spectral solar irradiance for clear sky conditions. We first consider the
theoretical results obtained for 10 different ice and snow surfaces with clear sky and
cloudy sky incident solar irradiance that show that the LCA spectral response may be
responsible for an overestimation of the theoretical albedo values by roughly 9% at
most. Then, the LCA values are compared with two "classical" albedometers over a
one-year measurement period (2013) for two sites in a tropical mountainous catchment
in Bolivia. One site is located on the Zongo Glacier (i.e. snow and ice surfaces) and
the second one is found on the right-hand side lateral moraine (bare soil and snow
surfaces). The results, at daily time steps (256 days), given by the LCA are in good
agreement with the classic albedo measurements taken with pyranometers with $R^2 =$
0.83 (RMSD = 0.10) and $R^2 = 0.92$ (RMSD = 0.08) for the Zongo Glacier and the right-
hand side lateral moraine, respectively. This demonstrates that our system performs
well and thus provides relevant opportunities to document spatio-temporal changes in
the surface albedo from direct observations at the scale of an entire catchment at a
low cost. Finally, during the period from September 2015 to June 2016, direct
observations were collected with 15 LCAs on the Zongo Glacier and successfully
compared with LANDSAT images showing the surface state of the glacier (i.e. snow



or ice). This comparison illustrates the efficiency of this system to monitor the daily
time step changes in the snow/ice coverage distributed on the glacier.
**Keywords:** Snow; Ice; Albedo; Glacier, Bolivia
**1-Introduction**
Albedo is a key variable controlling the surface energy balance through the shortwave
radiation budget. Documenting the spatio-temporal changes of this variable is a major
concern in hydrological modeling particularly in mountainous regions where the
seasonal snow and glacier covers induce significant and rapid changes in the surface
state with subsequent impacts on the energy budget. Hereafter, the spectral albedo is
defined as the ratio between the amount of energy reflected by the surface and the
incident energy for each wavelength of the solar spectrum (between 0.3 and 2.5 μm);
and the broadband albedo is the integration of the spectral albedo weighted by the
incident energy over the entire solar spectrum (0.3-2.5 μm). The amount of shortwave
radiation absorbed by the surface depends on the spectral and angular distribution of
the incident shortwave radiation and the surface characteristics, both of which are
highly variable in space and time (Stroeve *et al.*, 1997; Klok *et al.*, 2003). Clouds alter
the angular and spectral properties of the incident radiation. With respect to the snow
and ice surfaces, the albedo in the visible wavelength depends on the snow and ice
properties, the impurity amount (e.g. black carbon, dust, algae, etc.) and the snow
depth for the shallow snowpack. In the infrared portion of the spectrum, the albedo is
mainly controlled by the snow microstructure and is moderately sensitive to the solar
zenith angle (Warren, 1982). Liquid water and land have relatively low albedos (roughly
0.1 to 0.4) while snow and ice have higher albedos that typically can reach 0.9 for fresh
snow. It is still challenging to measure the temporal and spatial changes in the surface



albedo from the scale of specific points up to a regional scale. Different methods are
commonly used to retrieve albedo values from satellite images, ground photographs
or point measurements with pyranometers. Satellite-derived albedo maps provide
spatially continuous datasets but are limited to clear sky conditions; these maps may
contain significant uncertainties, especially over complex topographies (Stroeve *et al.*,
1997; Klok *et al.*, 2003; Dumont *et al.*, 2012), and provide averaged data over a pixel
size of hundreds of square meters. Ground photography using pairs of photographs in
the visible and infrared wavelengths is used to collect albedo maps that have a higher
spatial resolution than satellite images but which are limited by cloudy conditions, the
possible masking of the relief, an irregular grid due to the projection and more complex
ortho-rectification processes in mountainous regions (e.g. Corripio, 2004; Dumont *et*
*al.*, 2011). Finally, direct *in situ* snow and ice albedo measurements are sparse,
relatively expensive, often discontinuous and may contain large uncertainties if the
sensor is not regularly checked (Sicart *et al.*, 2001, van den Broeke *et al.*, 2004).
This article analyzes the efficiency of a low-cost albedometer (hereafter called LCA)
that measures the time series of *in-situ* incident and reflected illuminance values which
are used to calculate an accurate proxy of the albedo values called the *albedo index*.
The illuminance is the total luminous flux incident on a surface, per unit area. It is a
measure of how much the incident light illuminates the surface, wavelength-weighted
by the luminosity function to correlate with the human perception of brightness. In
section 2, we present the characteristics of and uncertainties on the LCA
measurements along with a comparison with the theoretical values for 10 different ice
and snow states and for two different incident irradiance spectra (cloudy or clear sky).
Then, section 3 presents two experiments carried out on a high-altitude tropical
mountain site in Bolivia (Zongo glacierized catchment). A first application for punctual



*in situ* measurements validates the LCA in the field via a comparison with classical
radiometers for two contrasting surfaces: snow/ice on the glacier or snow/bare soil on
the moraine. After that, a second application on the same glacier documents the
snow/ice changes on the surface of the glacier during the period that extends from
September 2015 to June 2016.
**2- LCA description and evaluation with theoretical albedo values for snow and**
**ice surfaces**
The LCA is comprised of two Hobo® Pendant Temperature/Light Data Loggers: one
for the incident illuminance and the other for the reflected illuminance (Fig. 1). The
sensor characteristics are given in Table 1. This sensor measures the illuminance in
lux and the measurement range is between 0 and 320,000 lux. The lux quantifies the
light incident flux per unit area. One lux equals one lumen per square meter with a
uniform distribution. In photometry, this unit is used as a measure of the intensity of
the light hitting or passing through a surface as perceived by the human eye. The
illuminance may be related to an energy quantified in watts per square meter ($W/m^2$),
but the conversion factor differs depending on the wavelength considered according to
the luminosity function, a standardized model of the human visual perception of
brightness. As a consequence, the illuminance depends on the spectral distribution of
the incident light.



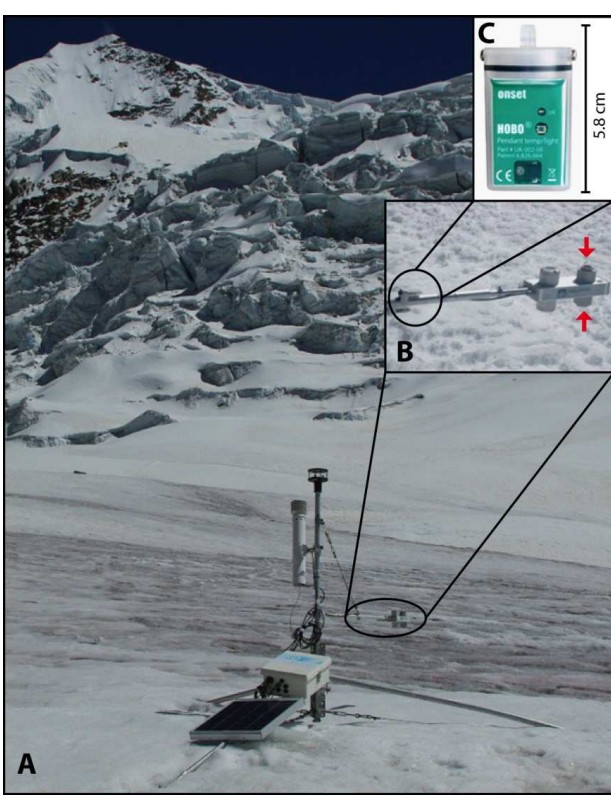


*Figure 1: A) Meteorological station on the Zongo Glacier; B) CNR1 radiometer (Kipp & Zonen) installed*
*at the SAMA meteorological station (the CM3 pyranometers are the two sensors on the right, red arrows)*
*and the LCA comprised of two Hobo® Pendant Temperature/Light Data Loggers (black arrows); C) Zoom*
*on a Hobo® Pendant Temperature/Light Data Logger (see Table 1 for detailed characteristics).*
The spectral range of the Hobo® Pendant Temperature/Light Data Logger is 0.3 to
1.195 µm (see Fig. 2). The spectral response of the sensor represents the amount of
incoming signal recorded by the sensor for any given wavelength and is reported in
Figure 2. Figure 2 shows that the spectral response of the sensor increases from 20 to
100% between 0.26 and 0.915 µm and then decreases until the upper limit of the
sensor sensitivity (i.e. 1.195 µm). The sensor detects roughly 85% of the total solar
irradiance for clear sky conditions (Fig. 2). Laboratory tests conducted with a
goniometer showed that the Hobo® Pendant Temperature/Light Data Logger cannot





measure the irradiance for incident zenith angles ranging from 55° to 90° (+/- 2°, where
0° is the vertical illumination). This is due to the design of the sensor (Fig. 1C).
Traditionally, the *in situ* albedo is measured using a CM3 pyranometer (Kipp & Zonen®)
in the shortwave domain from 0.305 to 2.800 µm (Fig. 1B). The CM3 is part of the
CNR1/CNR4 net radiometer, which is intended for the analysis of the radiation balance
of solar and thermal infrared radiation. The design of the CM3 is such that the upward-
facing and downward-facing sensors measure the energy received from the entire
hemisphere (a field of view of almost 180 degrees). The output is expressed in W/m².
The CM3 sensor has a 100% response for wavelengths between 0.305 to 2.8 µm (Fig.

129   2).

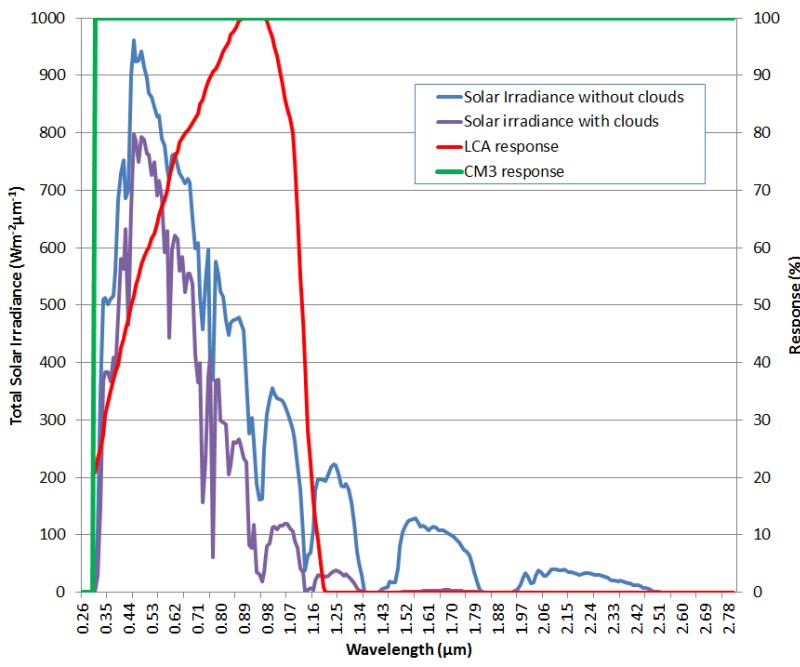



*Figure 2*: Hobo® Pendant Temperature/Light Data Logger and CM3 responses as a function of
*the wavelength and two examples of total solar irradiances for a clear sky in blue and for a cloudy sky*
*in purple given by the DISORT model (Stamnes et al., 1988) (Wm$^{-2}$μm$^{-1}$)*
*Table 1: Characteristics of the Hobo® Pendant Temperature/Light Data Logger sensor as per*
*the manufacturer*

| Measurement Range | Temperature: -20° to 70°C<br>Light: 0 to 320,000 lux |
|---|---|
| Accuracy | Temperature: +/- 0.53°C<br>Light intensity designed for measurement of relative light levels, see Figure 2 for the light wavelength response |
| Resolution | Temperature: 0.14°C at 25°C |
| Time accuracy | +/- 1 minute per month at 25°C |
| Operating range | in air: -20° to 70°C |
| Battery life | 1 year typical use |
| Memory | 64 K bytes |
| Material | Polypropylene case; stainless steel screws; Buna-N o-ring |
| Weight | 18 g |
| Dimensions | 58 x 33 x 23 mm |


It is noteworthy that the LCA contains an internal memory; this is not the case for the
CM3 pyranometers, which need to be connected to an external module for data
acquisition programming and data storage. The LCA cannot provide direct access to
the albedo as its response is not constant depending on the wavelength in the solar
spectrum. Finally, the conversion from illuminance to radiation in W/m² is not
straightforward since it depends on the spectral repartition of the incident and reflected
light.




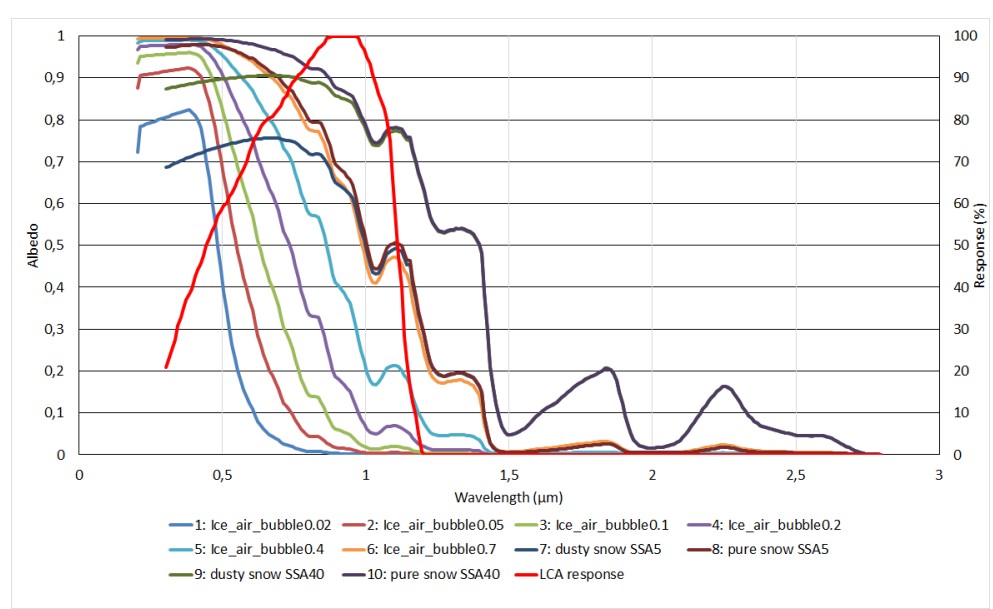


**Figure 3**: *Semi-infinite diffuse beam albedo of pure ice as a function of the effective air bubble radius*

*(mm) with a constant effective bubble concentration n'$_e$ = 0.3 mm$^{-3}$. Here 0.3 mm$^{-3}$ is the mean bubble*

*concentration determined from 28 Greenland and Antarctica ice core samples (Gardner and Sharp,*

*2010) - Semi-infinite diffuse beam albedo of dusty and pure snow from DISORT modelling with or without*

*dust and with a specific surface area (SSA) equal to 40 or 5 m$^2$ kg$^{-1}$ [Stamnes et al., 1988; Carmagnola*

*et al., 2013]. The dark green line shows the LCA response in %.*

Figure 3 shows 10 simulated spectral albedo curves for different glacier surfaces, four

for snow (with dusty or pure snow and with a specific surface area (SSA) equal to 5 or

40 m$^2$ kg$^{-1}$) and six for ice with different bubble concentrations (see Gardner and Sharp,

2010 for details). These 10 different surface types are used below to calculate the

theoretical uncertainty of the LCA measurements.

In the visible domain, the spectral albedo of pure snow is high (0.95) and the albedo

decreases in the infrared towards 0.1 for larger wavelengths (1.5-2 µm) (Fig. 3). For

dusty snow, the spectral albedo is lower than for pure snow. To calculate the



uncertainty for the ice covers, we chose pure ice that only contains air bubbles and no
impurity taken from the study of Gardner and Sharp (2010). In this case, all of the
photon absorption events will occur within the ice and all of the scattering will occur at
the ice-bubble boundaries, thereby neglecting all surface reflection as well as internal
scattering and absorption by the interstitial air (Mullen and Warren, 1988; Warren *et*
*al.*, 2002).
Two types of incident radiations are tested (clear sky and cloudy conditions given by
the SBDART model for the tropical Zongo latitude at 5000 m a.s.l., 23° solar zenith
angle, 0.1 atmospheric optical depth, (see Richiazzi et al., 1998 for details
concerning the model). The cloudy conditions are fully overcast with a cloud optical
depth equal to 64.
The theoretical broadband albedo and LCA *albedo indexes* are calculated over the
0.205-3.9 µm range using the theoretical solar irradiance and LCA spectral response
from Figure 2 and the semi-infinite diffuse beam albedo from Figure 3. The total
incident radiation flux for LCA is obtained by summing the theoretical incident radiation
fluxes weighted by the LCA response at each spectral increment of 5 microns, both for
cloudy and clear sky conditions. Similarly, the reflected radiation flux for the LCA is
obtained by summing the theoretical reflected radiation fluxes weighted by the LCA
response at each spectral increment of 5 µm, for each snow or ice class considered.
Then, the LCA *albedo index* is the ratio between the reflected and incident LCA
radiation fluxes, for each type of snow and ice surface and for cloudy or clear sky
conditions. Finally, this LCA *albedo index* is compared with the theoretical broadband
albedo when we consider the spectral variations. Note that the results are presented
with the incoming radiation corresponding to the total solar irradiances for clear sky


and cloudy sky conditions and without testing the effect of the angular limitation of the
LCA.






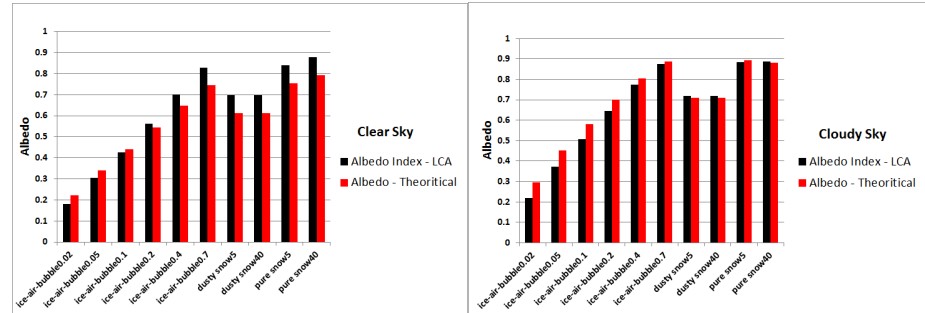

***Figure 4***: *Comparison between the theoretical semi-infinite diffuse beam broadband albedo*
*and albedo index calculated with the LCA for 10 different surfaces calculated with two kind of total solar*
*irradiance; on the right: cloudy sky and on the left: clear sky conditions (spectra are represented in Fig.*
*2) - 1: Ice air bubble 0.02; 2: Ice air bubble 0.05; 3: Ice air bubble 0.1; 4: Ice air bubble 0.2; 5: Ice air*
*bubble 0.4; 6: Ice air bubble 0.7; 7: dusty snow SSA 5 m² kg⁻¹; 8: dusty snow SSA 40 m² kg⁻¹; 9: pure*
*snow SSA 5 m² kg⁻¹; 10: pure snow SSA 40 m² kg⁻¹*
Figure 4 compares the theoretical albedos and the LCA *albedo index* with the
theoretical perfect albedo for the 10 surface configurations and for clear and cloudy
skies. Slight differences exist for ice with a bubble radius between 0.02 and 0.2 mm
with an underestimation of the LCA by 4% for a clear sky. For ice with an air bubble

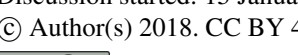



radius of 0.4 or 0.7 mm and for the two snow types (dusty and pure), the LCA tends to
overestimate the albedo by 8% in average for clear sky conditions. The LCA tends to
overestimate for albedo values higher than 0.5 (typically for snow) and to
underestimate for low values (i.e. for ice). A better agreement between the two sensors
is given in the cloudy case with an overall underestimation of 5% compared with 9%
for the clear sky case. This is explained by the response of the LCA based on the
wavelength, which is null for the 1.20-2.30 µm range (see Fig. 2).
**3-Applications on a high tropical glacierized catchment in Bolivia**
The Zongo Glacier (16°15′S, 68°10′W) is located in the Bolivian Cordillera Real (Fig.
5) between the Altiplano Plateau in the west and the Amazon Basin in the east. In
2006, the glacier covered an area of 1.96 km$^2$ extending from 6100 to 4900 m a.s.l.
(Rabatel *et al.*, 2012). The Bolivian Cordillera Real is located in the outer tropical zone,
which forms a transition zone between the tropics (continuously humid conditions) and
the subtropics (dry conditions). The climate of the outer tropics is characterized by low
seasonal temperature variability, high solar radiation influx all year round and marked
seasonal humidity and precipitation. The hydrological year (from September 1$^{st}$ to
August 31$^{st}$) can be divided into three periods: (1) September–December, with a
progressive increase in moisture and precipitation; (2) January–April, which is the core
period of the rainy season (approximately two-thirds of the total annual precipitation);
and (3) May–August, when dry conditions prevail (e.g. Sicart *et al.*, 2011). However,
precipitation can also occur during the dry period due to Southern Hemisphere mid-
latitude disturbances that track much further north of their usual path (e.g. Vuille and
Ammann, 1997; Sicart *et al.*, 2016).





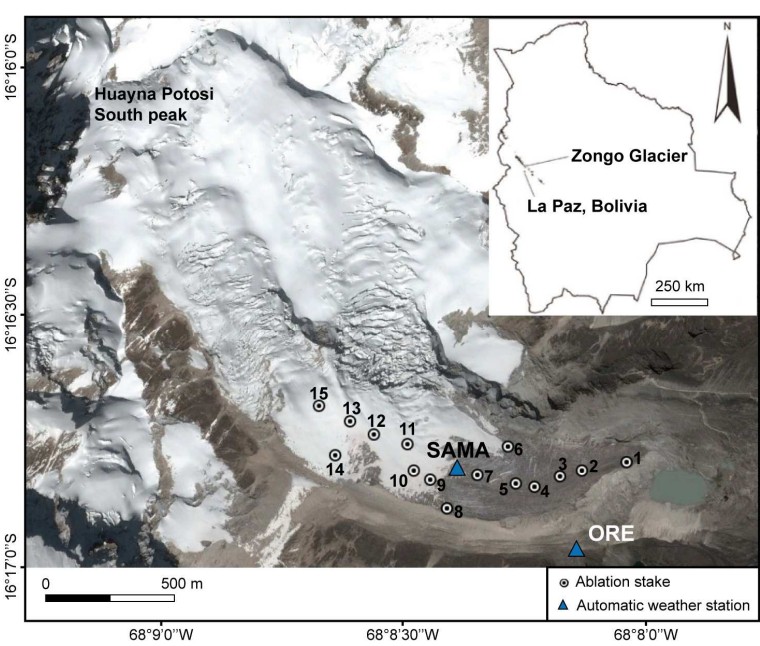


**Figure 5**: *Study site with the Zongo Glacier and the location of the meteorological stations: ORE (5050 m a.s.l.) outside of the glacier and SAMA (5056 m a.s.l.) on the glacier. The numbers indicate the position of each in situ LCA on ablation stakes.*

Two contrasting sites with different characteristics were chosen in order to evaluate the efficiency of the LCA (Figure 5). These two sites belong to the GLACIOCLIM observatory (https://glacioclim.osug.fr/) which has maintained a permanent glacio-meteo-hydrological monitoring program on the Zongo Glacier since 1991 (Rabatel *et al.*, 2013). The SAMA station is an automatic weather station (AWS) located on the Zongo Glacier (Figures. 1, 5) and the ORE station is a similar AWS located on the right-hand side lateral moraine. In order to capture the sky view for each station, ORE and SAMA, a digital elevation model (DEM) at 30-m resolution taken from ASTER images (Tachikawa *et al.*, 2011) was used. The sky view factor, which is the fraction of the celestial hemisphere visible from the surface defined by the local slope, was calculated with the SAGA GIS software (System for Automated Geoscientific Analyses,

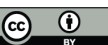



version 2.0.8) using the code provided by Boehner and Antonic (2009). The sky view
factors obtained are 0.92 and 0.98 for the SAMA and ORE stations, respectively.

242         Considering the limited field of view of the Hobo® Pendant Temperature/Light

Data Logger, daily albedo values are calculated between 11:00 AM and 3:00 PM local
time, ensuring that direct solar irradiance is caught by the two sensors. The *albedo*
*index* is calculated in two steps: (i) the sum of the hourly data for the incident
illuminance and the reflected illuminance between 11:00 AM and 3:00 PM; and (ii) the
calculation of the daily *albedo index* by dividing the reflected values by the incident
illuminance values. The time series used for the ORE and SAMA stations are
07/11/2012-06/03/2013 and 01/12/2012- 9/10/2013 respectively. Figures 6A and 6B
show the comparison between the CM3 albedo and LCA *albedo indexes* for the daily
values that range between 0.15 (dirty ice or bare soil) and 0.95 (fresh snow).

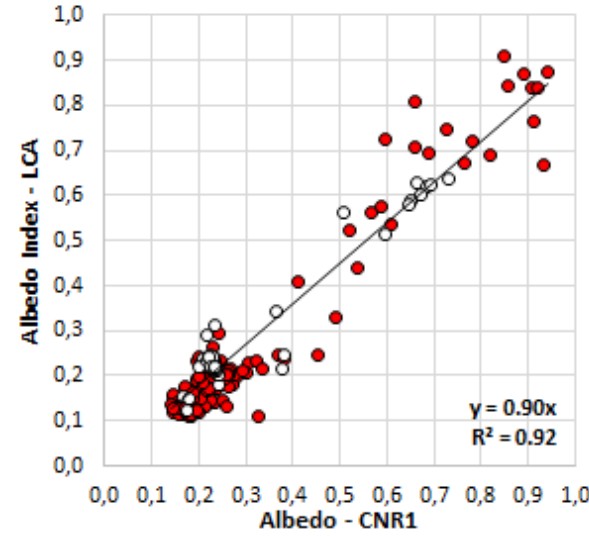

253         *A*



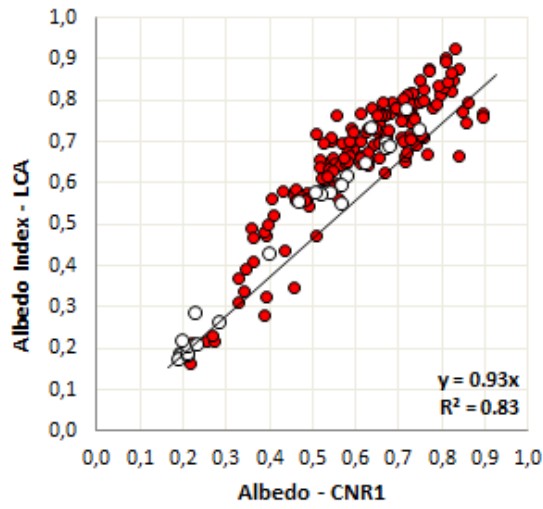

**B**
***Figure 6****: **A** Comparison of the daily measured albedo at the ORE site using the CNR1*
*radiometer and the LCA for the period from 07/11/2012-06/03/2013– daily data calculated for the 11 AM*
*– 3 PM time period – ORE; RMSD = 0.1; n = 263. **B** Comparison of the daily measured albedo at the*
*SAMA site on the Zongo Glacier using the CM3 sensor and LCA for the period from 01/12/2012-*
*9/10/2013 – daily data calculated for the 11 AM – 3 PM time period; RMSD = 0.08; n = 256. The red*
*dots are for cloudy conditions and the white dots are for sunny conditions, as per the classification given*
*by Sicart et al. (2016). The regression lines are calculated with all of the data.*
At the ORE site (Figure 6A), two groups of points can be distinguished. The lower
group (albedo close to 0.25) corresponds to measurements over bare soil. For the
second group, the broadband albedo and *albedo indexes* range from 0.3 to 0.9,
corresponding to several snow cover conditions: (i) thin and dirty snow; (ii)
homogeneous fresh snow; and (iii) patchy snow covers. There is good agreement
between the CM3 broadband albedos and LCA broadband *albedo indexes* ($R^2$ = 0.90
and RMSD = 0.08, with 256 days). The distribution for the albedos at the SAMA site
(Figure 6B) is more homogeneous. For the SAMA site, the albedo variations are due
to surface changes from ice to fresh snow. At this second site, there is also good



agreement between the CM3 and LCA albedo ($R^2$ = 0.93 and RMSD = 0.08, with 256
days).
The measurements are separated into two groups according to the sky conditions,
cloudy or sunny, as per the classification provided by Sicart et al. (2016). If we consider
the theoretical results from section 2, the LCA should give better results for cloudy
conditions; however there are not enough measurements for clear sky conditions
compared with the number of measurements for cloudy conditions to be able to come
to a conclusion. In both cases, the LCA tends to slightly overestimate the albedo values
by 5%. This result is in good agreement with the theoretical results presented in
Section 2 (Figure 4) showing that the LCA tends to overestimate the theoretical albedo
values for ice with bubbles and snow by less than 10%. The results are in good
agreement with the theoretical results obtained in section 2, with an overestimation for
the high albedos and an underestimation for the low albedos.
After the comparison between the CM3 and LCA, a second field experiment was
carried out in order to determine the spatio-temporal variability of the snow cover on
the Zongo Glacier during the period from 09/21/2015 to 06/30/2016. Fifteen LCA
stations were installed on ablation stakes distributed in the lower and middle part of
the glacier at altitudes ranging between 4929 and 5184 m a.s.l. (Figure 5). In order to
evaluate whether the LCA provides coherent information on the spatio-temporal
changes in the surface state of the glacier (fresh snow, old snow, ice), we compared
the LCA data with information retrieved from the LANDSAT images. With regards to
the LANDSAT images (30-m resolution), we first selected, within the archive, the cloud
free images recorded within the period when the LCA data were available (a list of the
23 images used here is provided in Table 2). On the LANDSAT images, we used a
spectral band combination involving the green, near-infrared (NIR) and middle infrared



(MIR) wavelengths (spectral bands # 2, 4 and 5 for LANDSAT images 5 and 7) which
is used to make a clear differentiation between snow and ice surfaces (Rabatel *et al.*,
2012). Then, according to the values in the NIR and MIR bands, the pixels where the
LCA are located were classified as snow covered (value of 2 in Figure 7) or ice covered
(value of 1 in Figure 7). In one case, the chosen value was 1.5 as the pixel showed
patchy snow cover. This can be explained if we consider that the spatial resolution of
the LANDSAT is equal to 900 $m^2$ and the surface view by the sensor is less than 1 $m^2$.
*Table 2: Date of the LANDSAT images used in the present study (Path/Row = 001/071)*
*(images from the web site: https://landsatlook.usgs.gov/viewer.html)*

| Date of the LANDSAT images | No. |
|---|---|
| 10/18/2015 | 1 |
| 11/03/2015 | 2 |
| 11/11/2015 | 3 |
| 11/19/2015 | 4 |
| 12/05/2015 | 5 |
| 12/13/2015 | 6 |
| 01/06/2016 | 7 |
| 01/14/2016 | 8 |
| 01/22/2016 | 9 |
| 02/15/2016 | 10 |
| 03/18/2016 | 11 |
| 03/26/2016 | 12 |
| 04/03/2016 | 13 |
| 04/11/2016 | 14 |
| 04/27/2016 | 15 |
| 05/13/2016 | 16 |
| 05/21/2016 | 17 |
| 05/29/2016 | 18 |
| 06/06/2016 | 19 |
| 06/14/2016 | 20 |
| 06/22/2016 | 21 |
| 06/30/2016 | 22 |
| 07/08/2016 | 23 |







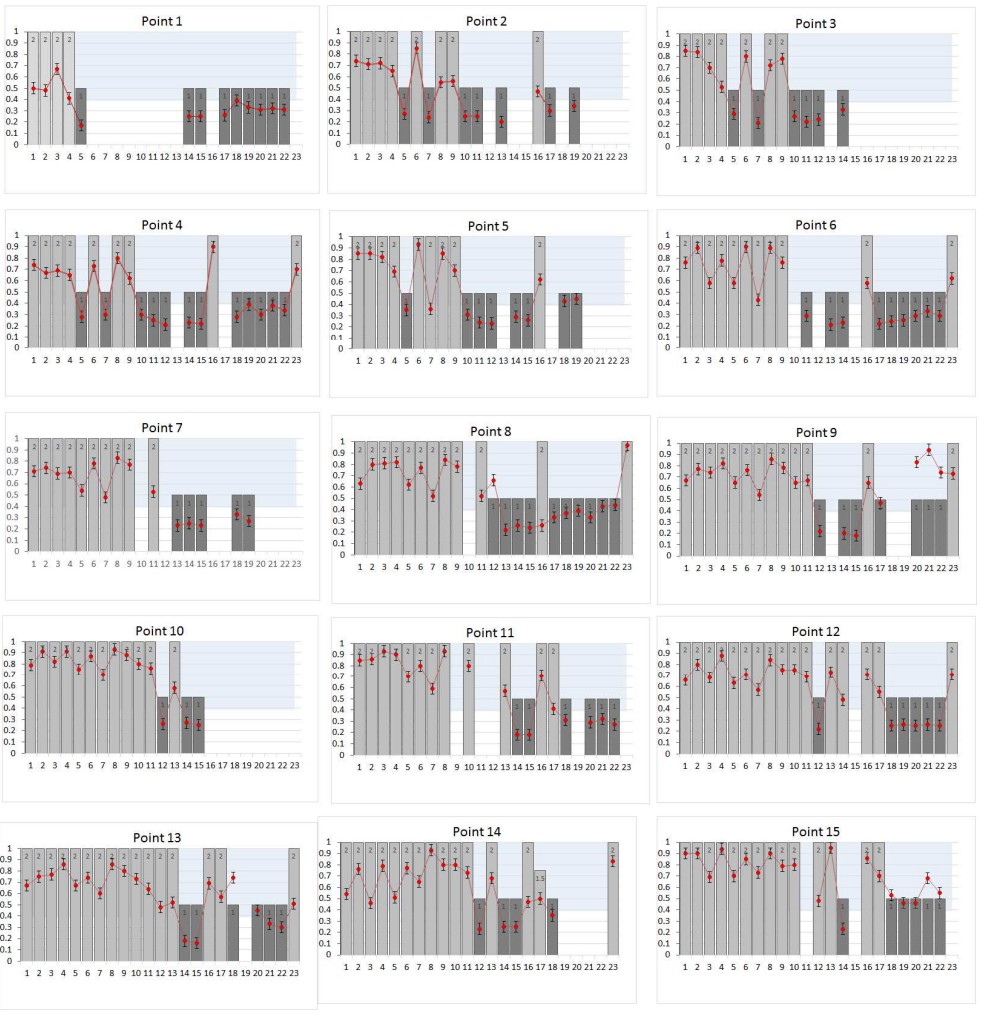


Figure 7: Comparison between the LCA measurements and the 23 LANDSAT images (from
10/18/2015 to 06/30/2016, the numbers for the Y axis are the image numbers, see Table 2 for the
correspondence) for the 15 points on the Zongo Glacier (see Figure 5 for the locations of the LCA). The
red points represent the albedo index value calculated with the LCA and the grey bars indicate the
surface state for the corresponding pixel (1: ice and 2: snow). A value of 1.5 was chosen for stake
number 14 as the pixel showed patchy snow cover.

The LCA network was deployed in the lower and middle part of the Zongo Glacier
(Figure 5) which is the zone where the snowline altitude goes up or down depending


on the snowfall events and ablation processes. For all of the points, we identified a first
period (10/18/2015 to 11/11/2015) with high albedo values comprised between 0.40
and 0.92. These values are in agreement with the surface state of the glacier on the
LANDSAT images where the pixels of the glacier tongue are all snow covered. During
the second period, the glacier surface is covered by ice or by snow depending on the
altitude. In further detail, we identified three groups organized by altitude ranges
depending on the changes in the surface state of the glacier with a first group in the
lower part of the glacier (LCA numbers 1, 2, 3, 4, 5), a second group in the middle part
of the glacier (LCA numbers 6, 7, 8, 9, 10, 11, 12) and a third group with LCA numbers
13, 14, 15 (see Figure 5 for the location). Finally, the comparisons between the *in situ*
LCA measurements and the surface state given by the LANDSAT images were used
to visually identify a threshold for the *albedo index* equal to 0.39 between snow and
ice. These results are in agreement with those obtained by Sicart *et al.* (2001) which
showed that the albedo for the Zongo Glacier ranges from 0.3 for dirty ice to 0.9 for
fresh snow.

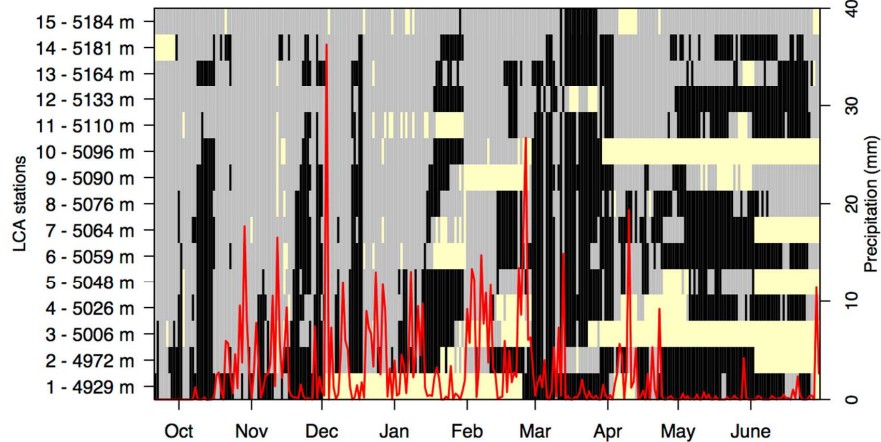




*Figure 8*: *Daily albedo index for the 15 LCA stations during the period from 09/21/2015 to 06/30/2016, in yellow: missing data; binary values considering the separation between ice (1: in black) and snow (2: in grey) with a threshold equal to 0.39. In red, the daily precipitation amount measured by the GEONOR rain gauge at the ORE station (mm/day).*

Figure 8 gives the evolution of the albedo for the 15 points during the period 09/21/2015-06/30/2016 as well as the precipitation amount measured by a GEONOR precipitation gauge at the ORE station (Figure 5). We can clearly identify the snowfall events and see how the snow disappears thus leaving the glacier ice exposed. As a result, the snowline altitude variations can be defined and vary between 4929 and 5184 m a.s.l. depending on the period of the year. In further detail, it can be noted that at the beginning of the study period (i.e. between September and November), the snowline quickly rises up and goes down due to intermittent precipitation events. Then, during the rainy season (from December to March), the glacier is mostly snow-covered (mainly above 5000 m a.s.l.). Finally, during the dry season (April to June), the snowline rises up to 5150 m a.s.l. and the glacier tongue is mainly snow free.

**4- Discussion and conclusion**

In this study we developed, evaluated and tested a new low-cost albedometer (LCA) comprised of two Hobo® Pendant Temperature/Light Data Loggers, measuring downward and upward illuminances. The measurements of the field of view of the LCA in the laboratory with a goniometer showed that the LCA cannot capture the radiation for zenith angles ranging from 55° to 90° (+/- 2°). Using the LCA spectral response (0.205 to 1.2 µm), we evaluated the simulated *albedo index* of the LCA over different types of snow and ice surfaces. The results showed that the LCA *albedo indexes* are within -4% to +8% of the theoretical broadband albedo values while considering that cloudy or clear sky incident irradiances only account only for the spectral response of



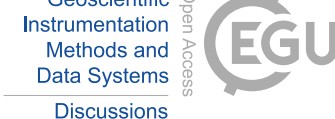

357 the LCA and not for the angular response of the LCA with respect to the ideal response.

358 In the second part of the study, we evaluated the LCA *albedo indexes* in the field using

359 CM3 broadband albedo values at two different sites in a tropical mountain in Bolivia:

360 on the Zongo Glacier, at one station located on the glacier and another one located on

361 the moraine. Data were recorded at hourly time steps and then the albedo indices were

362 calculated on a daily timescale (from 11:00 AM to 3:00 PM). The daily *albedo indexes*

363 from the LCA are in good agreement with the broadband albedo values derived from

364 the CM3 pyranometer. By comparing the LCA albedo estimates with the CM3

365 broadband albedo over a period of approximately 260 days at the two sites, we

366 conclude that the efficiency of the *albedo indexes* given by the LCA is +/- 0.1 compared

367 with classic CNR1 sensors. Future applications are certainly possible, especially

368 considering the low cost, the autonomy of the LCA in terms of energy and the very

369 small size of the sensors. For example, the LCA could be useful to spatialize *in situ*

370 albedos in glacierized catchments: both on the glacier, where the evolution of the snow

371 cover can be monitored, and in the non-glacierized part for the evolution of the

372 seasonal snow cover and, more generally, the changes in the ground albedo due to,

373 for example, variations in the soil moisture (Gascoin *et al.*, 2009). The comparison

374 between the LCA measurements and LANDSAT images during the period extending

375 from 10/18/2015 to 06/30/2016 showed that the LCA is a powerful tool that can be

376 used to quantify the evolution of the *albedo index* and to characterize the surface state

377 of the glacier by distinguishing between fresh snow, dirty snow and ice. In order to

378 have good results for the *albedo index* calculated with the LCA, a certain degree of

379 caution is required: for example, snow particles should not stay on the sensor and the

380 sensor must be kept horizontal. This new system has demonstrated its usefulness for

381 a tropical glacier and future studies in other climatic contexts should be conducted.



## 5- Acknowledgments

This study was funded by the French *Institut de Recherche pour le Développement* (IRD) through the Andean part of the French glacier observatory service, GLACIOCLIM (http://www-lgge.ujf-grenoble.fr/ServiceObs/SiteWebAndes/index.htm), and was carried out within the framework of the International Joint Laboratory GREAT-ICE, a joint initiative of the IRD as well as universities and institutions in Bolivia, Peru, Ecuador and Colombia. All of the contributing authors acknowledge the contribution of LABEX OSUG@2020, ANR grant No. ANR-10-LABX-56. The Pléiades satellite image used for Figure 1 was obtained from the CNES-ISIS FC18473 program funded by the BIOTHAW project (AAP-SCEN-2011-II). The authors would like to thank everyone who participated in the field campaigns: Patrick Ginot, Maxime Harter and Pierre Vincent.

## 8- References

Boehner, J., Antonic, O. (2009): 'Land-surface parameters specific to topo-climatology'. in: Hengl, T., Reuter, H. (Eds.): 'Geomorphometry - Concepts, Software, Applications'. Developments in Soil Science, Volume 33, p.195-226, Elsevier

Carmagnola, C. M., Domine, F., Dumont, M., Wright, P., Strellis, B., Bergin, M., et al. (2013). Snow spectral albedo at Summit, Greenland: comparison between in situ measurements and numerical simulations using measured physical and chemical properties of the snowpack. The Cryosphere, 7, 1139–1160. http://dx.doi.org/10.5194/tc-7-1139-2013.Colbeck, S.C., 1983, Theory of metamorphism of dry snow, Journal of Geophysical Research-Oceans and atmospheres, 88(NC9), 5475-5482



Corripio, J., 2004. Snow surface albedo estimation using terrestrial photography. Int.
J. Remote Sensing, 24(24), 5705-5729
Dumont, M., P. Sirguey, Y. Arnaud and D. Six. 2011. Monitoring spatial and temporal
variations of surface albedo on Saint Sorlin Glacier (French Alps) using terrestrial
photography. Cryosphere, 5, 759-771. doi: 10.5194/tc-5-759-2011
Dumont, M., J. Gardelle, P. Sirguey, A. Guillot, D. Six, A. Rabatel and Y. Arnaud.
2012. Linking glacier annual mass balance and glacier albedo retrieved from MODIS
data. Cryosphere, 6, 1527-1539. doi: 10.5194/tc-6-1527-2012
Gascoin, S., Ducharne, A., Ribstein, P., Perroy, E., Wagnon, P., 2009, Sensitivity of
bare soil albedo to surface soil moisture on the moraine of the Zongo glacier
(Bolivia), Geophysical Research Letters, volume 36, Issue 2,
DOI: 10.1029/2008GL036377
Gardner, A.S., Sharp, M.J., 2010, A review of snow and ice albedo and the
development of a new physically based broadband albedo parameterization,
Journal of Geophysical Research, VOL. 115, F01009, doi:10.1029/2009JF001444
Klok, E. J., W. Greuell, and J. Oerlemans (2003), Temporal and spatial variation of
the surface albedo of Morteratschgletscher, Switzerland, as derived from 12
Landsat images, J. Glaciol., 49, 491–502, doi:10.3189/172756503781830395.
Mullen, P.C, Warren, S.G., 1988, Theory of the optical properties of lake ice,
Volume 93, Issue D7, Pages 8403–8414DOI: 10.1029/JD093iD07p08403
Rabatel, A., A. Bermejo, E. Loarte, A. Soruco, J. Gomez, G. Leonardini, C. Vincent, J.-
E. Sicart. 2012. Can the snowline be used as an indicator of the equilibrium line and



mass balance for glaciers in the outer tropics? Journal of Glaciology, 58(212), 1027-
1036. doi: 10.3189/2012JoG12J027.
Rabatel, A., B. Francou, A. Soruco, J. Gomez, B. Caceres, J.L. Ceballos, R.
Basantes, M. Vuille, J.-E. Sicart, C. Huggel, M. Scheel, Y. Lejeune, Y. Arnaud, M.
Collet, T. Condom, G. Consoli, V. Favier, V. Jomelli, R. Galarraga, P. Ginot, L.
Maisincho, M. Ménégoz, J. Mendoza, E. Ramirez, P. Ribstein, W. Suarez, M.
Villacis, P. Wagnon. 2013. Current state of glaciers in the tropical Andes: a multi-
century perspective on glacier evolution and climate change. The Cryosphere, 7,
81-102. doi:10.5194/tc-7-81-2013.
Stamnes, K., Tsay, S.-C., Wiscombe, W., and Jayaweera, K.: Numerically stable
algorithm for discrete-ordinate-method radiative transfer in multiple scattering and
emitting layered media, Appl. Opt., 27, 2502–2509, 1988.
Sicart, J. E., P. Ribstein, P. Wagnon, and D. Brunstein (2001), Clear sky albedo
measurements on a sloping glacier surface. A case study in the Bolivian Andes,
Journal of Geophysical Research, 106(D23), 31729-31738
Sicart JE, Hock R, Ribstein P, Litt M and Ramirez E (2011) Analysis of seasonal
variations in mass balance and meltwater discharge of the tropical Zongo Glacier
by application of a distributed energy balance model. J. Geophys. Res., 116(D13),
D13105 (doi: 10.1029/2010JD015105)
Sicart J.E., Espinoza J.C., Quéno L. and M. Medina. (2016), Radiative properties of
clouds over a tropical Bolivian glacier: seasonal variations and relationship with
regional atmospheric circulation, International Journal of Climatology, Volume 36,
Issue 8, 3116–3128 (doi: 10.1002/joc.4540).



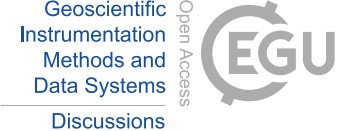

Stroeve, J., A. Nolin, and K. Steffen (1997), Comparison of AVHRRderived and in situ
surface albedo over the Greenland Ice Sheet, Remote Sens. Environ., 62, 262–
276, doi:10.1016/S0034-4257(97)00107-7.
Stamnes, K., Tsay, S.-C., Wiscombe, W., and Jayaweera, K., 1988, Numerically
stable algorithm for discrete-ordinate-methodradiative transfer in multiple
scattering and emitting layered media, Appl. Opt., 27, 2502–2509
Tachikawa, T., Kaku, M., Iwasaki, A., Gesch, D., Oimoen,M., Zhang, Z., Danielson,
J., Krieger, T., Curtis, B., Haase, J., Abrams,M., Crippen, R., Carabajal, C., 2011.
ASTER Global Digital Elevation Model Version 2 — Summary of validation results.
METI & NASA, (28 pp.).
van den Broeke, M., D. van As, C. Reijmer, and R. van de Wal, (2004), Assessing
and improving the quality of unattended radiation observations in Antarctica, J.
Atmos.   Oceanic   Technol.,   21,   1417–   1431,   doi:10.1175/1520-
0426(2004)021<1417:AAITQO>2.0.CO;2.
Vuille M and Ammann C (1997) Regional snowfall patterns in the high, arid Andes.
Climatic Change, 36(3–4), 413–423 (doi: 10.1023/A:1005330802974)
Warren, S.G., 1982, Optical-properties of snow, Reviews of Geophysics, 20(1), 67-89