# Peer review of "Discussion started: 15 January 2018 © Author(s) 2018. CC BY 4.0 License."

_Geoscientific Instrumentation, Methods and Data Systems, 2017_

## Referee Comment (RC1) · Anonymous Referee #1 · 21 Mar 2018

General comments

I have to say that I really enjoyed reading the paper. I believe testing new methods and techniques to reduce the gap in data availability on glacier energy balance is the way to go, specially because these relatively cheap instruments can support monitoring programs in many countries, where budgets are extremely limited and/or there are so many glaciers that even a large budget is not enough. Thus, I recommend this paper to be published. My (few) specific comments are intended for facilitating dissemination and correct interpretation of the findings from groups located in other places, not just

people working on the outer Tropics.

Specific comments

Page 7.L120. (also Page 14.L 243). Can you comment on the possible applicability of this instrument on calculating the same index in other regions where "hours" associated with these angles may be a little bit different? My impression is that the light sensor, although advertised as deployable in outdoors, was conceived for light measurements on more controlled environments, where the light source is between the limits described in the paper. I know that the authors suggest more studies at the end of the manuscript and I am not asking for a complete calculation of hours and days where this application would be ideal, but some notion of ranges could be good.

Table 1.- The same applies for the minimum operating range. There might be many locations where the -20°C is simply too high, as for example at the accumulation zones of mountain glaciers or whole glaciers in those located in sub-antarctic regions.

Page 14.L249 to Page 16.L283. and Fig 6. I wonder if it is possible to show the linear models for the individual land surfaces studies, i.e., bare soil and snow. My point here is that in 6A the cluster at the bottom (mostly bare soil?) might be influencing the slope for snow, which is the aim of the paper. In fact, reading the abstract (L.32) gives me the impression that this method works best for non-glacierized areas (rˆ2 0.83 versus 0.92). Another thing is perhaps including the error term in the equations of each plot; 6B seems to show a fairly consistent bias so perhaps in this case (for different snow conditions) a bias correction can improve the signal the authors are finding.

Aesthetic/Word choice suggestions

Page 2 L.26 and 31 (and other places where this word shows up), perhaps replace "Classical" for "Traditional"?

Page 2 L.29 and Page 13 L.235. To me "right hand side" is not a good way to refer to a location for the reader, because the point is actually located at the bottom of the map. I

suggest "the southern slope of the moraine arc (Fig 5)" or something along these lines.

Page 2. L.38. please reword "images showing the surface state of the glacier" (surface conditions?)

Page 5.L88, I suggest deleting the word "classical"

Fig. 1: I really don't see the black arrows.

Page 6.L112, the abstract says 0.26 instead of 0.3

Page 6.L115 "Figure 2" twice (even with a dot in between) difficults the flow of the document to me. I suggest changing the second "Figure 2" for "In that figure" or something similar.

Page 8. L142. I don't think "repartition" is the right word, perhaps "distribution"

Page 10.L166. I don't see where this parenthesis closes.

Page 10. L169-170. Suggest replacing "a cloud optical depth equal to 64" for "an optical depth of 64".

Page 10. L173-178, Wouldn't this explanation be clearer using equations instead?

Figure 4. It says theoritical instead of theoretical. Also, I think this figure is too small.

Page 20. L336-346. I feel this paragraph is a bit disconnected from all the previous text. I see no previous reference on snowline elevation or to precipitation behavior. Perhaps they need to reference figure 8 in this paragraph.

---

## Referee Comment (RC2) · Anonymous Referee #2 · 25 Mar 2018

General Comments:

The need for sub-pixel scale measurements of albedo is clear and the authors demonstrate a cost effective approach to albedo measurements with some limitations, but broader potential for applications over snow, ice and other surfaces. The article provides a theoretical synopsis to compare spectral response of the HOBO Pendant Light logger and typical response of various snow and ice surfaces. There is acknowledgement of the limited field of view and hence restricting the measurements to times of day when the solar elevation angle is relatively high (perhaps 55 degrees or higher between

11am and 3pm). Results of the measurements at 15 locations along an elevation transect over the glacier reveal interesting and valuable results along with comparison to satellite imagery of ice-snow surface character. Onset Computer has a good reputation for cost-effective sensors and loggers and the Pendant loggers are easily/quickly obtained from various Internet vendors.

Specific Comments:

The article would benefit from more background evidence of in situ snow and ice albedo measurement studies in the past, such as P. 4 L74-76 needs expanded. Maybe elaborate on previous techniques: Brock, B., Willis, I., & Sharp, M. (2000). Measurement and parameterization of albedo variations at Haut Glacier d'Arolla, Switzerland. Journal of Glaciology, 46(155), 675-688. doi:10.3189/172756500781832675.

The authors provide a comparison between simi-infinite diffuse albedo and the albedo index computed with the LCA for 10 different surface conditions, but it must be made clear that Figure 4 is comparing theory to theory, not theory to measurements by the LCA, rather it is a comparison to the expected albedo index based on the spectral response of the surface and the LCA.

If the authors had the means and resources, it would be much more convincing and valuable to compare the theoretical semi-infinte diffuse bean broadband albedo to actual measurements by the LCA over the 10 surface types in a controlled lab environment.

Nevertheless, HOBO Pendant Loggers are robust, reliable and easily installed parallel to the horizon, so this technique is accessible to a broad audience for embedded field observations. At less than $70. USD each, it's very cost effective.

Technical Corrections Suggestions:

Hobo should be all upper case HOBO.

P2, L26: please describe the "classical" albedometer; it would be good if you could

demonstrate that this is a secondary standard for albedo measurement or solar irradiance measurement, as this makes your comparison study more reliable.

P2, L29: remove "right-hand side" and be more descriptive in terms of slope and azimuth direction of this location

P9, L158: for _longer_ wavelengths

P10, L171: Is there a formula to illustrate the calculation of the theoretical LCA albedo that clarifies the method?

P11, L 193: calculated with the LCA is vague. Do you mean theoretically estimated based on spectral response of the LCA?

P12, L205: What two sensors? There are no sensors involved (now direct measurements) in the theoretical estimations. Sensors are used in section 3.

P14, L247: dividing the _sum_ of reflected by _sum_ of incident?

P18, L309: numbers for the X axis, not Y axis?

P21, L379: be a little more explicit about the precautions that we need to consider when applying the Hobo albedometer

---

## Author Comment (AC1) · 29 Apr 2018

We are grateful to the anonymous reviewer for his positive feedbacks. We present below our detailed answer to each of their points. The reviewer's comments appear in black Times font and our responses appear in brown Arial font. Page 7.L120. (also Page 14.L 243). Can you comment on the possible applicability of this instrument on calculating the same index in other regions where "hours" associated with these angles may be a little bit different? My impression is that the light sensor, although advertised as deployable in outdoors, was conceived for light measurements on more controlled environments, where the light source is between the limits described in the paper. I know that the authors suggest more studies at the end of the manuscript and I am not asking for a complete calculation of hours and days where this application would be ideal, but some notion of ranges could be good.

The angle of view of the sensor is 55°; which limits where and when it can be used. To determine these limits, we calculated what the solar angle is at noon for different latitudes throughout the year. Considering the LCA is operational when the solar angle is greater than 55° at noon, it may be used all year long at latitudes between 12°N and 12°S, from March to October between 12°N and 30°N, and from September to March between 12°S and 30°S. The sensor cannot be used at latitudes higher than 60°N or 60°S at any time throughout the year. Between 45°N and 45°S the sensors can be operated during the ablation season when the glacier surface changes are the most important.

We specify the applicability of the LCA in terms of latitudes and periods of the year in the revised version of our manuscript ('Abstract' and 'Discussion and conclusion' sections).

In the abstract you can now read: 'Despite the limits imposed by the angle view restrictions, the LCA can be used between 45°N and 45°S during the ablation season (spring and summer) when the melt rate related to the albedo is the most important.'

In the Discussion and conclusion we specify the applicability of the LCA as follows: 'The angle of view of the sensor is 55°; which limits where and when it can be used. To determine these limits, we calculated what the solar angle is at noon for different latitudes throughout the year. Considering the LCA is operational when the solar angle is greater than 55° at noon, it may be used all year long at latitudes between 12°N and 12°S, from March to October between 12°N and 30°N, and from September to March between 12°S and 30°S. The sensor cannot be used at latitudes higher than 60°N or 60°S at any time throughout the year. Between 45°N and 45°S the sensors can be operated during the ablation season when the glacier surface changes are the most important.'

Table 1.- The same applies for the minimum operating range. There might be many locations where the -20_C is simply too high, as for example at the accumulation zones of mountain glaciers or whole glaciers in those located in sub-antarctic regions. This is true and might be a limitation for the application of the LCA in very cold regions. Therefore, we added a sentence that indicates this in the revised version as follows: 'Due to its operating temperature range (see table 1), the use of the LCA is limited at very cold locations where the temperature falls continuously below -20°C for long periods of time. However, this may not be too critical since the main purpose of the device is to document albedo surface changes during melt periods when such low temperature conditions are not typical.'

Page 14.L249 to Page 16.L283. and Fig 6. I wonder if it is possible to show the linear models for the individual land surfaces studies, i.e., bare soil and snow. My point here is that in 6A the cluster at the bottom (mostly bare soil?) might be influencing the slope for snow, which is the aim of the paper. In fact, reading the abstract (L.32) gives me the impression that this method works best for non-glacierized areas (rËĘ2 0.83 versus 0.92). Another thing is perhaps including the error term in the equations of each plot; 6B seems to show a fairly consistent bias so perhaps in this case (for different snow conditions) a bias correction can improve the signal the authors are finding. We made a new version of figure 6 according to your comments: we included different regression lines considering all of the data; under cloudy and sunny conditions. The caption of the Figure 6 now reads: "Figure 6: A Comparison of the daily measured albedo at the ORE site using the CNR1 radiometer and the LCA for the period from 07/11/2012 to 06/03/2013 – daily data calculated from 11AM to 3PM – ORE; RMSD = 0.1; n = 247. B Comparison of the daily measured albedo at the SAMA site on the Zongo Glacier using the CM3 sensor and LCA for the period from 01/12/2012 to 9/10/2013 – daily data calculated from 11AM to 3PM; RMSD = 0.08; n = 175. The red dots are for cloudy conditions and the white dots are for sunny conditions, as per the classification given by Sicart et al. (2016). The calculated regression lines are shown in red for cloudy conditions, blue for sunny conditions, and black for all conditions. The dotted lines represent the bisectors."

Page 2 L.26 and 31 (and other places where this word shows up), perhaps replace "Classical" for "Traditional"? We agree and replaced "Classical" with "Traditional".

Page 2 L.29 and Page 13 L.235. To me "right hand side" is not a good way to refer to a location for the reader, because the point is actually located at the bottom of the map. I suggest "the southern slope of the moraine arc (Fig 5)" or something along these lines. In order to be more explicit, the text has been modified according to your remark as follows: "on the crest of the lateral moraine".

Page 2. L.38. please reword "images showing the surface state of the glacier" (surface conditions?) We reworded "images showing the surface state of the glacier (i.e snow or ice)" by "images showing the surface conditions of the glacier (i.e. snow or ice)."

Page 5.L88, I suggest deleting the word "classical" We changed the word "classical" to "traditional".

Fig. 1: I really don't see the black arrows. You are right. We modified the figure by adding the black arrows.

Page 6.L112, the abstract says 0.26 instead of 0.3 We changed the value in the text from 0.3 to 0.26.

Page 6.L115 "Figure 2" twice (even with a dot in between) difficults the flow of the document to me. I suggest changing the second "Figure 2" for "In that figure" or something similar. As the referee suggested, we changed "Figure 2" to "This figure".

Page 8. L142. I don't think "repartition" is the right word, perhaps "distribution" We changed the word "repartition" to "distribution".

Page 10.L166. I don't see where this parenthesis closes. We closed the parenthesis.

Page 10. L169-170. Suggest replacing "a cloud optical depth equal to 64" for "an optical depth of 64". As suggested, we changed the wording "a cloud optical depth equal to 64" to "an optical depth of 64".

Page 10. L173-178, Wouldn't this explanation be clearer using equations instead? We agree with this comment and added equations in order to be clearer. In the revised version you can now read: "The theoretical broadband albedo and LCA albedo indexes are calculated over the 0.205-3.9 $\mu$m range using the theoretical solar irradiance, the LCA spectral response from Figure 2, and the semi-infinite diffuse beam albedo from Figure 3. The total incident radiation flux for LCA, Sinc (in W m-2), is obtained by summing the theoretical incident radiation fluxes, Sinc-th($\lambda$) (in W m-2 $\mu$m-1), weighted by the LCA response, R$\lambda$ (-), at each spectral increment of 5 $\mu$m for both cloudy and clear sky conditions (Eq. 1). S_inc=$\sum$_($\lambda$=0.205)ˆ3.9âŰŠãĂŰS_(inc-th) ($\lambda$) R_$\lambda$ d$\lambda$ãĂŮ (Eq. 1) Similarly, the reflected radiation flux for the LCA, Sref (in W m-2), is obtained by summing the theoretical reflected radiation fluxes, Sref-th($\lambda$) (in W m-2 $\mu$m-1), weighted by the LCA response, R$\lambda$ (-), at each spectral increment of 5 $\mu$m for each snow or ice class considered (Eq. 2). S_ref=$\sum$_($\lambda$=0.205)ˆ3.9âŰŠãĂŰS_(ref-th) ($\lambda$) R_$\lambda$ d$\lambda$ãĂŮ (Eq. 2) Then, the LCA albedo index, Albedoindex (-),is the ratio between the reflected and incident LCA radiation fluxes for each type of snow and ice surface, and cloudy or clear sky conditions (Eq. 3). ãĂŰAlbedoãĂŮ_index=S_ref/S_inc (Eq. 3)."

Figure 4. It says theoritical instead of theoretical. Also, I think this figure is too small. We changed "theoritical" to "theoretical" and enlarged the two figures.

Page 20. L336-346. I feel this paragraph is a bit disconnected from all the previous text. I see no previous reference on snowline elevation or to precipitation behavior. Perhaps they need to reference figure 8 in this paragraph. We agree with this comment and described the dynamic of the snowline elevation in section 3. You can now read: 'For the whole glacier, the main precipitation type is solid and the albedo increases after each snowfall with a snowline that could reach the front of the glacier. After that, during dry consecutive days the snowline rises up due to the snow melting processes.' We added the following sentence to give the logical chain between the two sections: 'Using this threshold, it is possible to plot the evolution of the glacier cover (even ice or snow) over time for different altitudes ranging from 4929 m a.s.l. to 5184 m a.s.l. (figure 8).'

You will find the revised manuscript in supplement.

Please also note the supplement to this comment:
https://www.geosci-instrum-method-data-syst-discuss.net/gi-2017-55/gi-2017-55-AC1-supplement.pdf

**Supplement:**

**Submitted to: Geoscientific Instrumentation, Methods and Data Systems (GI)**

**Technical note: A low-cost albedometer for snow and ice measurements – Theoretical results and application on a tropical mountain in Bolivia**

Thomas Condom[1][*], Marie Dumont[2], Lise Mourre[1], Jean Emmanuel Sicart[1], Antoine Rabatel[1], Alessandra Viani[1], Alvaro Soruco[3]

[1] Université de Grenoble Alpes, IRD, CNRS, Grenoble-INP, IGE (UMR5001), F-38000 Grenoble, France

[2] Météo-France, CNRS, CNRM-GAME/CEN (UMR3589), Grenoble, France

[3] UMSA, Instituto de Geológicas y del Medio Ambiente, La Paz, Bolivia

*Corresponding author: thomas.condom@ird.fr

**Abstract**

This study presents a new instrument called a low-cost albedometer (LCA) composed of two illuminance sensors that are used to measure *in-situ* incident and reflected illuminance values on a daily timescale. The ratio between reflected *vs.* incident illuminances is called the *albedo index* and can be compared with actual albedo values. Due to the shape of the sensor, the direct radiation for zenith angles ranging from 55° to 90° is not measured. The spectral response of the LCA varies with the solar irradiance wavelengths within the range 0.26 to 1.195 μm, and the LCA detects 85% of the total spectral solar irradiance for clear sky conditions. We first consider the theoretical results obtained for 10 different ice and snow surfaces with clear sky and cloudy sky incident solar irradiance that show that the LCA spectral response may be responsible for an overestimation of the theoretical albedo values by roughly 9% at most. Then, the LCA values are compared with two "traditional" albedometers CM3 pyranometer (Kipp & Zonen®) in the shortwave domain from 0.305 to 2.800 μm over a one-year measurement period (2013) for two sites in a tropical mountainous catchment in Bolivia. One site is located on the Zongo Glacier (i.e. snow and ice surfaces) and the second one is found on the crest of the lateral moraine (bare soil and snow surfaces) which present a horizontal surface and a sky view factor of 0.98. The results, at daily time steps (256 days), given by the LCA are in good agreement with the classic albedo measurements taken with pyranometers with $R^2 = 0.83$ (RMSD = 0.10) and $R^2 = 0.92$ (RMSD = 0.08) for the Zongo Glacier and the right-hand side lateral moraine, respectively. This demonstrates that our system performs well and thus provides relevant opportunities to document spatio-temporal changes in the surface albedo from direct observations at the scale of an entire catchment at a low cost. Finally, during the period from September 2015 to June 2016, direct observations were collected with 15

LCAs on the Zongo Glacier and successfully compared with LANDSAT images showing the surface conditions of the glacier (i.e snow or ice). This comparison illustrates the efficiency of this system to monitor the daily time step changes in the snow/ice coverage distributed on the glacier. Despite the limits imposed by the angle view restrictions, the LCA can be used between 45°N and 45°S during the ablation season (spring and summer) when the melt rate related to the albedo is the most important.

**Keywords:** Snow; Ice; Albedo; Glacier, Bolivia

**1-Introduction**

Albedo is a key variable controlling the surface energy balance through the shortwave radiation budget. Documenting the spatio-temporal changes of this variable is a major concern in hydrological modeling particularly in mountainous regions where the seasonal snow and glacier covers induce significant and rapid changes in the surface state with subsequent impacts on the energy budget. Hereafter, the spectral albedo is defined as the ratio between the amount of energy reflected by the surface and the incident energy for each wavelength of the solar spectrum (between 0.3 and 2.5 µm); and the broadband albedo is the integration of the spectral albedo weighted by the incident energy over the entire solar spectrum (0.3-2.5 µm). The amount of shortwave radiation absorbed by the surface depends on the spectral and angular distribution of the incident shortwave radiation and the surface characteristics, both of which are highly variable in space and time (Stroeve *et al.*, 1997; Klok *et al.*, 2003). Clouds alter the angular and spectral properties of the incident radiation. With respect to the snow and ice surfaces, the albedo in the visible wavelength depends on the snow and ice properties, the impurity amount (e.g. black carbon, dust, algae, etc.) and the snow depth for the shallow snowpack. In the infrared portion of the spectrum, the albedo is mainly controlled by the snow microstructure and is moderately sensitive to the solar zenith angle (Warren, 1982). Liquid water and land have relatively low albedos (roughly 0.1 to 0.4) while snow and ice have higher albedos that typically can reach 0.9 for fresh snow. It is still challenging to measure the temporal and spatial changes in the surface albedo from the scale of specific points up to a regional scale. Different methods are commonly used to retrieve albedo values from satellite images, ground photographs or point measurements with pyranometers. Satellite-derived albedo maps provide spatially continuous datasets but are limited to clear sky conditions; these maps may contain significant uncertainties, especially over complex topographies (Stroeve *et al.*, 1997; Klok *et al.*, 2003; Dumont *et al.*, 2012), and provide averaged data over a pixel size of hundreds of square meters. Ground photography using pairs of photographs in the visible and infrared wavelengths is used to collect albedo maps that have a higher spatial resolution than satellite images but which are limited by cloudy conditions, the possible masking of the relief, an irregular grid due to the projection and more complex ortho-rectification processes in mountainous regions (e.g. Corripio, 2004; Dumont *et al.*, 2011). Finally, direct *in situ* snow and ice albedo measurements are sparse, relatively expensive, often discontinuous and may contain large uncertainties if the sensor is not regularly checked (Sicart *et al.*, 2001, van den Broeke *et al.*, 2004).

A study published by Brock et al. (2000) aimed to document the spatial and temporal variations of surface albedo on the Haut Glacier d'Arolla, Swizerland during the 1993 and 1994 ablation seasons (from the mid-May to the end of August). They used traditional Kipp and Zonen CM7B albedometer (that is expensive) and relied the temporal variations of albedo with surface conditions as snow depth, surface snow density and surface snow grain-size. One of their conclusions underlined the importance to conduct in-situ field measurements continuously at daily time scale across a glacier throughout the ablation season, as the measurements are crucial to develop albedo parametrization into hydro-glaciological models.

This article analyzes the efficiency of a low-cost albedometer (hereafter called LCA) that measures the time series of *in-situ* incident and reflected illuminance values which are used to calculate an accurate proxy of the albedo values called the *albedo index*. The illuminance is the total luminous flux incident on a surface, per unit area. It is a measure of how much the incident light illuminates the surface, wavelength-weighted by the luminosity function to correlate with the human perception of brightness. In section 2, we present the characteristics of and uncertainties on the LCA measurements along with a comparison with the theoretical values for 10 different ice and snow states and for two different incident irradiance spectra (cloudy or clear sky). Then, section 3 presents two experiments carried out on a high-altitude tropical mountain site in Bolivia (Zongo glacierized catchment). A first application for punctual *in situ* measurements validates the LCA in the field via a comparison with traditional radiometers for two contrasting surfaces: snow/ice on the glacier or snow/bare soil on the moraine. After that, a second application on the same glacier documents the snow/ice changes on the surface of the glacier during the period that extends from September 2015 to June 2016.

**2- LCA description and evaluation with theoretical albedo values for snow and ice surfaces**

The LCA is comprised of two HOBO® Pendant Temperature/Light Data Loggers: one for the incident illuminance and the other for the reflected illuminance (Fig. 1). The sensor characteristics are given in Table 1. This sensor measures the illuminance in lux and the measurement range is between 0 and 320,000 lux. The lux quantifies the light incident flux per unit area. One lux equals one lumen per square meter with a uniform distribution. In photometry, this unit is used as a measure of the intensity of the light hitting or passing through a surface as perceived by the human eye. The illuminance may be related to an energy quantified in watts per square meter ($W/m^2$), but the conversion factor differs depending on the wavelength considered according to the luminosity function, a standardized model of the human visual perception of brightness. As a consequence, the illuminance depends on the spectral distribution of the incident light. Due to its operating temperature range (see table 1), the use of the LCA is limited at very cold locations where the temperature falls continuously below -20°C for long periods of time. However, this may not be too critical since the main purpose of the device is to document albedo surface changes during melt periods when such low temperature conditions are not typical.

[Figure]

**_Figure 1_**_: A) Meteorological station on the Zongo Glacier; B) CNR1 radiometer (Kipp & Zonen) installed_

_at the SAMA meteorological station (the CM3 pyranometers are the two sensors on the right, red arrows)_

_and the LCA comprised of two HOBO® Pendant Temperature/Light Data Loggers (black arrows); C)_

_Zoom on a HOBO® Pendant Temperature/Light Data Logger (see Table 1 for detailed characteristics)._

The spectral range of the HOBO® Pendant Temperature/Light Data Logger is 0.26 to

1.195 µm (see Fig. 2). The spectral response of the sensor represents the amount of incoming signal recorded by the sensor for any given wavelength and is reported in

Figure 2. This figure shows that the spectral response of the sensor increases from 20

to 100% between 0.26 and 0.915 µm and then decreases until the upper limit of the sensor sensitivity (i.e. 1.195 µm). The sensor detects roughly 85% of the total solar irradiance for clear sky conditions (Fig. 2). Laboratory tests conducted with a goniometer showed that the HOBO® Pendant Temperature/Light Data Logger cannot measure the irradiance for incident zenith angles ranging from 55° to 90° (+/- 2°, where

0° is the vertical illumination). This is due to the design of the sensor (Fig. 1C).

Traditionally, the *in situ* albedo is measured using a CM3 pyranometer (Kipp & Zonen®)

in the shortwave domain from 0.305 to 2.800 µm (Fig. 1B). The CM3 is part of the

CNR1/CNR4 net radiometer, which is intended for the analysis of the radiation balance of solar and thermal infrared radiation. The design of the CM3 is such that the upward- facing and downward-facing sensors measure the energy received from the entire hemisphere (a field of view of almost 180 degrees). The output is expressed in W/m$^2$.

The CM3 sensor has a 100% response for wavelengths between 0.305 to 2.8 µm (Fig.

2).

[Figure]

*Figure 2: HOBO® Pendant Temperature/Light Data Logger and CM3 responses as a function*

*of the wavelength and two examples of total solar irradiances for a clear sky in blue and for a cloudy sky*

*in purple given by the DISORT model (Stamnes et al., 1988) ($Wm^{-2}\mu m^{-1}$)*

*Table 1: Characteristics of the HOBO® Pendant Temperature/Light Data Logger sensor as per*

*the manufacturer*

| Measurement | Temperature: -20° to 70°C |
| --- | --- |
| Range | Light: 0 to 320,000 lux |
| Accuracy | Temperature: +/- 0.53°C
Light intensity designed for measurement of relative light levels, see Figure 2 for the light wavelength response |
| Resolution | Temperature: 0.14°C at 25°C |
| Time accuracy | +/- 1 minute per month at 25°C |
| Operating range | in air: -20° to 70°C |
| Battery life | 1 year typical use |
| Memory | 64 K bytes |
| Material | Polypropylene case; stainless steel screws; Buna-N o-ring |
| Weight | 18 g |
| Dimensions | 58 x 33 x 23 mm |

It is noteworthy that the LCA contains an internal memory; this is not the case for the

CM3 pyranometers, which need to be connected to an external module for data acquisition programming and data storage. The LCA cannot provide direct access to the albedo as its response is not constant depending on the wavelength in the solar spectrum. Finally, the conversion from illuminance to radiation in W/m² is not straightforward since it depends on the spectral distribution of the incident and reflected light.

[Figure]

**Figure 3**: *Semi-infinite diffuse beam albedo of pure ice as a function of the effective air bubble radius*

*(mm) with a constant effective bubble concentration $n'_e = 0.3\ mm^{-3}$. Here $0.3\ mm^{-3}$ is the mean bubble*

*concentration determined from 28 Greenland and Antarctica ice core samples (Gardner and Sharp,*

*2010) - Semi-infinite diffuse beam albedo of dusty and pure snow from DISORT modelling with or without*

*dust and with a specific surface area (SSA) equal to 40 or 5 $m^2\ kg^{-1}$ [Stamnes et al., 1988; Carmagnola*

*et al., 2013]. The red line shows the LCA response in %.*

Figure 3 shows 10 simulated spectral albedo curves for different glacier surfaces, four

for snow (with dusty or pure snow and with a specific surface area (SSA) equal to 5 or

$m^2\ kg^{-1}$) and six for ice with different bubble concentrations (see Gardner and Sharp,

for details). These 10 different surface types are used below to calculate the

theoretical uncertainty of the LCA measurements.

In the visible domain, the spectral albedo of pure snow is high (0.95) and the albedo

decreases in the infrared towards 0.1 for longer wavelengths (1.5-2 µm) (Fig. 3). For

dusty snow, the spectral albedo is lower than for pure snow. To calculate the

uncertainty for the ice covers, we chose pure ice that only contains air bubbles and no

impurity taken from the study of Gardner and Sharp (2010). In this case, all of the photon absorption events will occur within the ice and all of the scattering will occur at the ice-bubble boundaries, thereby neglecting all surface reflection as well as internal scattering and absorption by the interstitial air (Mullen and Warren, 1988; Warren *et al.*, 2002).

Two types of incident radiations are tested (clear sky and cloudy conditions given by the SBDART model for the tropical Zongo latitude at 5000 m a.s.l., 23° solar zenith angle, 0.1 atmospheric optical depth), (see Richiazzi et al., 1998 for details concerning the model). The cloudy conditions are fully overcast with an optical depth of 64.

The theoretical broadband albedo and LCA *albedo indexes* are calculated over the 0.205-3.9 µm range using the theoretical solar irradiance, the LCA spectral response from Figure 2, and the semi-infinite diffuse beam albedo from Figure 3. The total incident radiation flux for LCA, $S_{inc}$ (in W m$^{-2}$), is obtained by summing the theoretical incident radiation fluxes, $S_{inc\text{-}th}(\lambda)$ (in W m$^{-2}$ µm$^{-1}$), weighted by the LCA response, $R_\lambda$ (-), at each spectral increment of 5 µm for both cloudy and clear sky conditions (Eq. 1).

$$S_{inc} = \sum_{\lambda=0.205}^{3.9} S_{inc-th}(\lambda)R_\lambda d\lambda \qquad\qquad \text{(Eq. 1)}$$

Similarly, the reflected radiation flux for the LCA, $S_{ref}$ (in W m$^{-2}$), is obtained by summing the theoretical reflected radiation fluxes, $S_{ref\text{-}th}(\lambda)$ (in W m$^{-2}$ µm$^{-1}$), weighted by the LCA response, $R_\lambda$ (-), at each spectral increment of 5 µm for each snow or ice class considered (Eq. 2).

$$S_{ref} = \sum_{\lambda=0.205}^{3.9} S_{ref-th}(\lambda)R_\lambda d\lambda \qquad\qquad \text{(Eq. 2)}$$

Then, the LCA *albedo index, Albedo$_{index}$* (-),is the ratio between the reflected and incident LCA radiation fluxes for each type of snow and ice surface and for cloudy or clear sky conditions (Eq. 3).

$$Albedo_{index} = \frac{S_{ref}}{S_{inc}}$$ (Eq. 3)

Finally, this LCA *albedo index* is compared with the theoretical broadband albedo when we consider the spectral variations. Note that the results are presented with the incoming radiation corresponding to the total solar irradiances for clear sky and cloudy sky conditions and without testing the effect of the angular limitation of the LCA.

[Figure]

[Figure]

*Figure 4*: *Comparison between the theoretical semi-infinite diffuse beam broadband albedo and LCA albedo index theoretically estimated based on spectral response of the LCA for 10 different surfaces calculated with two kinds of total solar irradiance (see the text for the calculation); on the right: cloudy sky and on the left: clear sky conditions (spectra are represented in Fig. 2) - 1: Ice air bubble 0.02; 2: Ice air bubble 0.05; 3: Ice air bubble 0.1; 4: Ice air bubble 0.2; 5: Ice air bubble 0.4; 6: Ice air bubble 0.7; 7: dusty snow SSA 5 $m^2$ $kg^{-1}$; 8: dusty snow SSA 40 $m^2$ $kg^{-1}$; 9: pure snow SSA 5 $m^2$ $kg^{-1}$; 10: pure snow SSA 40 $m^2$ $kg^{-1}$*

Figure 4 compares the theoretical albedos and the LCA *albedo index* with the theoretical perfect albedo for the 10 surface configurations and for clear and cloudy skies. Slight differences exist for ice with a bubble radius between 0.02 and 0.2 mm with an underestimation of the LCA by 4% for a clear sky. For ice with an air bubble radius of 0.4 or 0.7 mm and for the two snow types (dusty and pure), the LCA tends to overestimate the albedo by 8% in average for clear sky conditions. The LCA tends to overestimate for albedo values higher than 0.5 (typically for snow) and to underestimate for low values (i.e. for ice). A better agreement between the theoretical albedos and the LCA *albedo index* is given in the cloudy case with an overall underestimation of 5% compared with 9% for the clear sky case. This is explained by the response of the LCA based on the wavelength, which is null for the 1.20-2.30 μm range (see Fig. 2).

**3-Applications on a high tropical glacierized catchment in Bolivia**

The Zongo Glacier (16°15′S, 68°10′W) is located in the Bolivian Cordillera Real (Fig. 5) between the Altiplano Plateau in the west and the Amazon Basin in the east. In 2006, the glacier covered an area of 1.96 km$^2$ extending from 6100 to 4900 m a.s.l. (Rabatel *et al.*, 2012). For the whole glacier, the main precipitation type is solid and the albedo increases after each snowfall with a snowline that could reach the front of the glacier. After that, during dry consecutive days the snowline rises up due to the snow melting processes. The Bolivian Cordillera Real is located in the outer tropical zone, which forms a transition zone between the tropics (continuously humid conditions) and the subtropics (dry conditions). The climate of the outer tropics is characterized by low seasonal temperature variability, high solar radiation influx all year round and marked seasonal humidity and precipitation. The hydrological year (from September 1$^{st}$ to August 31$^{st}$) can be divided into three periods: (1) September–December, with a progressive increase in moisture and precipitation; (2) January–April, which is the core period of the rainy season (approximately two-thirds of the total annual precipitation); and (3) May–August, when dry conditions prevail (e.g. Sicart *et al.*, 2011). However, precipitation can also occur during the dry period due to Southern Hemisphere mid-latitude disturbances that track much further north of their usual path (e.g. Vuille and Ammann, 1997; Sicart *et al.*, 2016).

[Figure]

**Figure 5**: *Study site with the Zongo Glacier and the location of the meteorological stations: ORE (5050 m a.s.l.) outside of the glacier and SAMA (5056 m a.s.l.) on the glacier. The numbers indicate the position of each in situ LCA on ablation stakes.*

Two contrasting sites with different characteristics were chosen in order to evaluate the efficiency of the LCA (Figure 5). These two sites belong to the GLACIOCLIM observatory (https://glacioclim.osug.fr/) which has maintained a permanent glacio-meteo-hydrological monitoring program on the Zongo Glacier since 1991 (Rabatel *et al.*, 2013). The SAMA station is an automatic weather station (AWS) located on the Zongo Glacier (Figures. 1, 5) and the ORE station is a similar AWS located on the crest of the lateral moraine. In order to capture the sky view for each station, ORE and SAMA, a digital elevation model (DEM) at 30-m resolution taken from ASTER images (Tachikawa *et al.*, 2011) was used. The sky view factor, which is the fraction of the celestial hemisphere visible from the surface defined by the local slope, was calculated with the SAGA GIS software (System for Automated Geoscientific Analyses, version 2.0.8) using the code provided by Boehner and Antonic (2009). The sky view factors obtained are 0.92 and 0.98 for the SAMA and ORE stations, respectively.

Considering the limited field of view of the HOBO® Pendant Temperature/Light

Data Logger, daily albedo values are calculated between 11:00 AM and 3:00 PM local time, ensuring that direct solar irradiance is caught by the two sensors. The *albedo*

*index* is calculated in two steps: (i) the sum of the hourly data for the incident illuminance and the reflected illuminance between 11:00 AM and 3:00 PM; and (ii) the calculation of the daily *albedo index* by dividing the sum of reflected values by the sum of incident illuminance values. The time series used for the ORE and SAMA stations are 07/11/2012-06/03/2013 and 01/12/2012- 9/10/2013 respectively. Figures 6A and

6B show the comparison between the CM3 albedo and LCA *albedo indexes* for the daily values that range between 0.15 (dirty ice or bare soil) and 0.95 (fresh snow).

[Figure]

*Figure 6: A Comparison of the daily measured albedo at the ORE site using the CNR1 radiometer and*

*the LCA for the period from 07/11/2012 to 06/03/2013 – daily data calculated from 11AM to 3PM – ORE;*

*RMSD = 0.1; n = 247. **B** Comparison of the daily measured albedo at the SAMA site on the Zongo*

*Glacier using the CM3 sensor and LCA for the period from 01/12/2012 to 9/10/2013 – daily data*

*calculated from 11AM to 3PM; RMSD = 0.08; n = 175. The red dots are for cloudy conditions and the*

*white dots are for sunny conditions, as per the classification given by Sicart et al. (2016). The calculated*

*regression lines are shown in red for cloudy conditions, blue for sunny conditions, and black for all*

*conditions. The dotted lines represent the bisectors.*

[revised manuscript text omitted]

**4- Discussion and conclusion**

In this study we developed, evaluated and tested a new low-cost albedometer (LCA) comprised of two HOBO® Pendant Temperature/Light Data Loggers, measuring downward and upward illuminances. The measurements of the field of view of the LCA in the laboratory with a goniometer showed that the LCA cannot capture the radiation for zenith angles ranging from 55° to 90° (+/- 2°). The angle of view of the sensor is 55°; which limits where and when it can be used. To determine these limits, we calculated what the solar angle is at noon for different latitudes throughout the year. Considering the LCA is operational when the solar angle is greater than 55° at noon, it may be used all year long at latitudes between 12°N and 12°S, from March to October between 12°N and 30°N, and from September to March between 12°S and 30°S. The sensor cannot be used at latitudes higher than 60°N or 60°S at any time throughout the year. Between 45°N and 45°S the sensors can be operated during the ablation season when the glacier surface changes are the most important. Using the LCA spectral response (0.205 to 1.2 µm), we evaluated the simulated *albedo index* of the LCA over different types of snow and ice surfaces. The results showed that the LCA *albedo indexes* are within -4% to +8% of the theoretical broadband albedo values while considering that cloudy or clear sky incident irradiances only account only for the spectral response of the LCA and not for the angular response of the LCA with respect to the ideal response. In the second part of the study, we evaluated the LCA *albedo indexes* in the field using CM3 broadband albedo values at two different sites in a tropical mountain in Bolivia: on the Zongo Glacier, at one station located on the glacier and another one located on the moraine. Data were recorded at hourly time steps and then the albedo indices were calculated on a daily timescale (from 11:00 AM to 3:00 PM). The daily *albedo indexes* from the LCA are in good agreement with the broadband albedo values derived from the CM3 pyranometer. By comparing the LCA albedo estimates with the CM3 broadband albedo over a period of approximately 260 days at the two sites, we conclude that the efficiency of the *albedo indexes* given by the LCA is +/- 0.1 compared with classic CNR1 sensors. Future applications are certainly possible, especially considering the low cost, the autonomy of the LCA in terms of energy and the very small size of the sensors. For example, the LCA could be useful to spatialize *in situ* albedos in glacierized catchments: both on the glacier, where the evolution of the snow cover can be monitored, and in the non-glacierized part for the evolution of the seasonal snow cover and, more generally, the changes in the ground albedo due to, for example, variations in the soil moisture (Gascoin *et al.*, 2009). The comparison between the LCA measurements and LANDSAT images during the period extending from 10/18/2015 to 06/30/2016 showed that the LCA is a powerful tool that can be used to quantify the evolution of the *albedo index* and to characterize the surface state of the glacier by distinguishing between fresh snow, dirty snow and ice. In order to have good results for the *albedo index* calculated with the LCA, a certain degree of caution is required: for example, snow particles should not stay on the sensor and the sensor must be kept horizontal. Therefore, we recommend a frequency of about 15 days between each field visit and data download. This new system has demonstrated its usefulness for a tropical glacier and future studies in other climatic contexts should be conducted.

**5- Acknowledgments**

This study was funded by the French *Institut de Recherche pour le Développement* (IRD) through the Andean part of the French glacier observatory service, GLACIOCLIM (http://www-lgge.ujf-grenoble.fr/ServiceObs/SiteWebAndes/index.htm), and was carried out within the framework of the International Joint Laboratory GREAT-ICE, a joint initiative of the IRD as well as universities and institutions in Bolivia, Peru, Ecuador and Colombia. All of the contributing authors acknowledge the contribution of LABEX OSUG@2020, ANR grant No. ANR-10-LABX-56. The Pléiades satellite image used for Figure 1 was obtained from the CNES-ISIS FC18473 program funded by the BIOTHAW project (AAP-SCEN-2011-II). The authors would like to thank everyone who participated in the field campaigns: Patrick Ginot, Maxime Harter and Pierre Vincent. We thank Sara Mullin and Lance Brooks for the correction of English text.

**8- References**

Boehner, J., Antonic, O. (2009): 'Land-surface parameters specific to topo-climatology'. in: Hengl, T., Reuter, H. (Eds.): 'Geomorphometry - Concepts, Software, Applications'. Developments in Soil Science, Volume 33, p.195-226, Elsevier

Brock, B., Willis, I., Sharp, M. (2000). Measurement and parametrization of albedo variations at Haut Glacier d'arolla, Swizerland. Journal of Glaciology, 46(155), 675-688. Doi:10.3189/172756500781832675

[revised manuscript text omitted]